# COVID-19 Vaccine-Induced Lichenoid Eruptions—Clinical and Histopathologic Spectrum in a Case Series of Fifteen Patients with Review of the Literature

**DOI:** 10.3390/vaccines11020438

**Published:** 2023-02-14

**Authors:** Yonatan K. Sapadin, Elazar Mermelstein, Robert G. Phelps, Christopher F. Basler, JoAnn M. Tufariello, Mark G. Lebwohl

**Affiliations:** 1Stevenson University, Owings Mills, MD 21117, USA; 2Yeshiva University, New York, NY 10033, USA; 3Department of Pathology, Icahn School of Medicine at Mount Sinai, One Gustave L. Levy Place, New York, NY 10029, USA; 4Department of Dermatology, Icahn School of Medicine at Mount Sinai, New York, NY 10029, USA; 5Department of Microbiology, Icahn School of Medicine at Mount Sinai, New York, NY 10029, USA; 6Clinical Therapeutics, Icahn School of Medicine at Mount Sinai, New York, NY 10029, USA

**Keywords:** vaccine, vaccination, COVID-19, lichen planus, lichenoid eruption, lichenoid reaction, Blaschkoid, autoimmune, lines of Blasschko, BLAISE

## Abstract

Lichen planus is a distinctive mucocutaneous disease with well-established clinical and histopathologic criteria. Lichenoid eruptions closely resemble lichen planus and may sometimes be indistinguishable from it. Systemic agents previously associated have included medications, viral infections and vaccines. Sporadic case reports of lichen planus and lichenoid reactions associated with COVID-19 vaccines have recently emerged. Herein, we review the world literature (31 patients) and expand it with a case series of 15 patients who presented with vaccine-induced lichenoid eruption (V-ILE). The spectrum of clinical and histopathologic findings is discussed with emphasis on the subset whose lesions manifested in embryologic fusion lines termed lines of Blaschko. This rare Blaschkoid distribution appeared in seven of the 46 patients studied. Of interest, all seven were linked to the mRNA COVID-19 vaccines. We believe that all lichenoid eruptions should be approached with a heightened index of suspicion and patients should be specifically questioned with regards to their vaccination history. When diagnosed early in its course, V-ILE is easily treated and resolves quickly in almost all patients with or without hyperpigmentation. Additional investigative studies regarding its immunopathology and inflammatory signaling pathways may offer insight into other Th1-driven autoimmune phenomena related to COVID-19 vaccination.

## 1. Introduction

A novel coronavirus disease (COVID-19) caused by the severe respiratory syndrome coronavirus 2 (SARS-CoV-2) emerged in China in December 2019 and spread quickly around the globe. COVID-19 was declared a pandemic several months later by the World Health Organization (WHO). In December 2020, the United States Food and Drug Administration granted emergency use authorization for the messenger RNA (mRNA)-1273 (Moderna) and BNT162b2 (Pfizer-BioNTech) vaccines. Campaigns to combat the devastation from COVID-19 were initiated in Europe, the United States and soon afterwards, in other countries around the world. In February 2021, a similar approval for the adenoviral vector vaccine Ad26.COV2.S (J&J/Janssen) was issued.

Patients in this study received one of three COVID-19 vaccines produced by Moderna, Pfizer-BioNTech or J&J/Janssen. The Pfizer-BioNTech and Moderna vaccines are lipid nanoparticles that contain an N1-methyl-pseudouridine (m1Ψ) nucleoside-modified mRNA that encodes the SARS-CoV-2 spike glycoprotein (S). The m1Ψ modification functions to suppress the immunogenicity and enhance the stability of the RNA [1,2].The J&J/Janssen vaccine, in contrast, is a recombinant, replication-incompetent adenovirus type 26 vector that encodes S. In all three vaccines, the S protein is mutated to stabilize it in its pre-fusion conformation [3] in order to promote the display of neutralizing epitopes and to improve expression [4,5]. Each of these vaccines elicits a broad immune system stimulation, including anti-S antibody and cell mediated responses [6,7,8,9,10,11]. Neutralizing and non-neutralizing antibodies are produced, as well as antibodies with Fc-mediated effector functions, such as antibody-dependent (AD) neutrophil phagocytosis, AD monocyte cellular phagocytosis, AD complement deposition and AD natural killer cell activation [11,12].

These vaccines also trigger anti-S T cell responses in animal models and in humans that include T_H_1-biased CD4 responses. These are characterized by the enhanced production of IFN-γ, IL-2 and TNF-α, cytokines that are thought to play a central role in the immunopathogenesis of lichen planus. This subject was coherently reviewed and illustrated in a 2009 review article by Sontheimer [13]. CD4+ Th1 T cells secrete proinflammatory cytokines, including IFN-γ and IL-2, which bind to receptors on CD8+ cytotoxic T cells. These cytotoxic effector cells express FasL and secrete granzymes, perforin and TNF-α–, all of which have been theorized to contribute to the induction of basal cell keratinocyte apoptosis [13]. Other studies emphasize granule exocytosis (with the release of perforin and Granzyme B) and not the Fas/Fas-ligand system as the primary CD4+ and CD8+ T lymphocyte—mediated cytotoxic pathway in humans [14]. 

These robust adaptive immune responses reflect the activation of antigen presenting cells (APCs). The plasmacytoid dendritic cell (pDC), one type of professional APC, is involved in the early events of this inflammatory cascade [15]. It may be theorized that in the early stages of a COVID-19 V-ILE plasmacytoid dendritic cells, through MHC class II molecules, present S to CD4+ T cells which become activated and stimulate a Th1-biased response. Additionally, the secretion of type 1 IFNs, including IFN-α, from activated pDCs and keratinocytes may play a crucial role in directing the recruitment and migration of CD4+ and CD8+ effector T lymphocytes from the circulation to the sites of lesional skin. This perpetuates a cycle of T cell-mediated epidermal damage in lichenoid eruptions which continues until other factors supervene to establish remission. Additional detail is provided in the more recent article by Tziotzios et al. [16]. 

Data from reports of reactions to the vaccines has been and is continuing to be accumulated from real-world evidence. Localized reactions at the intramuscular injection site are very common and include erythema, pain, swelling and a delayed hypersensitivity reaction dubbed “COVID arm” [17] or “COVID vaccine arm” [18]. A broad spectrum of other reactions and eruptions in the skin may occur (Figure 1). Reports of these reactions initially trickled into the literature but have since been more firmly established. One such reaction is termed lichenoid eruption (LE) because it mimics or may be identical to lichen planus (LP) clinically and histologically. To discuss the features of LEs, it is therefore important to first review the features of its prototype, LP itself.

LP, a distinctive mucocutaneous disease with well-established clinical and histopathologic criteria, is mediated by complex immunologic events. It is conveniently referred to as the “disease of the Ps” because several of its prominent clinical features begin with the letter “P”. The cutaneous eruption is characterized by pruritic purple, planar (flat-topped), polygonal papules and plaques (raised lesions) with a predilection for the periphery (flexural aspects of the arms and legs).

Blaschkoid LP is a rare variant of LP in which the lesions are distributed in the lines of Blaschko, an invisible pattern of lines on the skin formed during embryogenesis (Figure 2). This series of lines was first elucidated by Alfred Blaschko, a German dermatologist, in 1901 at the 7th Congress of the German Dermatological Society in Breslau [19]. These lines, normally invisible, may become noticeable in many skin disorders. These include specific genetic syndromes and congenital or acquired skin diseases. 

LP has a stereotypical pathology. The epidermis shows varying degrees of acanthosis and orthokeratosis. Wedge-shaped hypergranulosis is characteristic. At the dermoepidermal junction, there are foci of vacuolar alteration and apoptotic (necrotic) keratinocytes. These are caused by the underlying infiltrate which is confined to the superficial (papillary) dermis below and effaces, in part, the overlying epidermis. Its configuration is bandlike, and it is composed of lymphocytes and histiocytes.

By contrast, exogenous agents, such as drugs and vaccines, may induce simulations of the pathology described above and produce a lichenoid tissue reaction, an atypical pattern that often, but not always, shows variations from the classic features. Clues may appear in the epidermis or in the distribution or types of inflammatory cells present in the infiltrate. The epidermis can show parakeratosis and spongiosis. Apoptotic keratinocytes and lymphocytes may be numerous and seen at all levels of the epidermis. The infiltrate in the dermis is focally bandlike but can additionally be patchy and may involve the mid or deeper dermis. It is composed of lymphocytes and histiocytes, but other cell types may include eosinophils, plasma cells and even epithelioid cells. When the epidermis shows papillomatosis or elongate rete, an underlying keratosis or lentigo may be present and this lesion, often solitary, is called a lichenoid keratosis.

For the discussion in this paper, the histopathologic reaction is termed lichen planus only when it shows the typical pattern. Despite bearing some overlapping pathologic features, a reaction showing any alteration to or deviation from this classic pattern is atypical and the impression is termed lichenoid dermatitis. This diagnosis may prompt an inquiry from the pathologist to the clinician regarding the differential diagnosis of potential triggers including exogenous agents. The umbrella term for the clinical diagnosis which correlates to this pathology is lichenoid eruption (LE). When the implicated exogenous agent is a drug, the term lichenoid drug eruption (LDE) may be used. Analogously, when the suspected agent is a vaccine, we propose that the more specific descriptive term, vaccine-induced lichenoid eruption (V-ILE), should be used.

Lichen striatus (LS), a cutaneous eruption usually seen in children, may also show lichenoid dermatitis on histopathologic examination. It is the prototype of acquired inflammatory dermatoses that appear along Blaschko’s lines. A retrospective study analyzing 115 affected children found an association with atopy in 70 cases [20]. It most commonly presents with asymptomatic or slightly itchy, small, skin-colored to erythematous papules that may coalesce on the extremities.

Rarely, a LS-like eruption may present in adults and may manifest in a different anatomic location, such as the trunk. These eruptions show a different histopathology termed spongiotic dermatitis and some authors refer to these cases as “adult Blaschkitis” [21]. Because inflammatory dermatoses that occur in the lines of Blaschko, such as lichen striatus, “linear” lichen planus and adult Blaschkitis, may overlap and share clinical and histopathologic features, the unifying term “Blaschkolinear acquired inflammatory skin eruptions”, referred to by the acronym BLAISE, has been proposed by some authors to capture all of these cases under one descriptive heading [22]. This latter term is preferentially used here for those vaccine-induced eruptions that were distributed in the lines of Blaschko and that showed lichenoid dermatitis on skin biopsy, whether or not all of the stereotypical histopathologic criteria were met for a clear diagnosis of an inflammatory skin disease, such as lichen planus or lichen striatus.

While the etiology of LP has not yet been completely elucidated, systemic agents associated with LP and other lichenoid tissue reactions include medications, viral infections and vaccines [23,24,25,26,27,28,29,30,31,32,33]. Herein, the literature reporting lichenoid eruptions linked to COVID-19 vaccines is reviewed. We expand on the spectrum of the clinical and histopathologic findings of these authors by reporting a series of an additional fifteen cases and compare the data from these two sets of cases. Theories regarding immunopathogenesis including molecular mimicry, a potential pathomechanism that may explain how vaccination in a susceptible host could trigger a lichenoid eruption, are discussed.

## 2. Materials and Methods

### 2.1. Study Design

#### 2.1.1. COVID-19 VILE: Case Series of Fifteen Patients

Background: New Jersey recorded its first COVID-19 case on 4 March 2020. In December 2020, the United States Food and Drug Administration approved the worldwide use of messenger RNA (mRNA)-based vaccines, Pfizer/BioNTech (BNT162b2) and Moderna (mRNA-1273) to address the pandemic. On 15 December, New Jersey began its vaccination campaign to counterattack the virus, which had already killed more than 17,000 New Jerseyans and 300,000 people nationwide. Non-healthcare workers began to receive their first COVID-19 vaccination one month later, on 15 January 2021.

Over a 14-month period, from April 2021 to June 2022, patients who presented for evaluation of a lichenoid eruption and whose diagnosis based on skin biopsy was either lichen planus or lichenoid dermatitis were studied. Employing a standardized template, a detailed medical history was systematically taken from all patients included for analysis. To be a candidate for this study, the interval between the date of the patient’s COVID-19 vaccination and the onset of the cutaneous eruption was 3 weeks or less. Patients whose intervals were longer were excluded. The patients’ vaccination cards were used to document the dates of the vaccinations and boosters.

None of the 15 patients had a history of hepatitis C or hepatitis B. One patient tested positive for COVID-19 3 months prior to receiving his first COVID-19 vaccine. The first lesions of his eruption appeared 2 weeks after his second vaccine. None of the patients had a recently added medication or a recent change in the dosage of their medications. Two patients had a longstanding history of lichen planus, one with oral lichen planus and one with cutaneous lichen planus. In each case, the last prior episode was 15 years ago or longer prior to reactivation.

Fifteen patients—thirteen at an outpatient dermatology practice in Hackensack, New Jersey and two at the Mount Sinai Medical Center dermatology clinic in New York qualified for this study. Informed consent was obtained from each patient in his or her own language to perform skin biopsies to confirm the clinical diagnosis. A media consent form was signed by each of the patients to publish any identifying material in online and print publications. After the diagnoses were confirmed by pathology, the patients were treated and followed longitudinally for their clinical course and to determine if subsequent vaccinations induced additional eruptions. The fifteen patients were divided into two groups based on whether their eruption followed the lines of Blaschko or followed a random non-Blaschkoid distribution (Figure 3).

#### 2.1.2. Review of the Literature

##### Search Strategy

Publications were searched for relevant papers employing Pubmed and Scopus databases. The beginning of the time period searched was December 2019, when the first reports of SARS-CoV-2 emerged. The end point was 21 October 2022. The following terms in varying combinations were used for the search: “COVID-19”, “Vaccine”, “Vaccination”, “Lichen Planus”, “Lichenoid Eruption”, “Lichenoid Reaction”, “Vaccine-Induced”, “Vaccine-Triggered”, “Vaccine-Related”, “Vaccine-Associated”, “Lichenoid Drug Eruption”, “Lichenoid Dermatitis”. Additionally considered for inclusion were the references and citations listed in the resulting articles. Eligible articles had no language restrictions. Case reports and case series (largest reported four patients) of patients were included if the criteria for clinical and histopathologic diagnosis were deemed to have been met in the report.

The following data were extracted from the included publications: Author, Year of Publication, Type of Vaccine/Manufacturer, Type of Eruption (reactivation versus new-onset), Gender, Race, Time to Onset of Eruption, Time to Onset of Eruption after first dose of vaccine, Time to Onset after second dose of vaccine, Eruptions after Successive Vaccinations, Treatment and Clinical Course. These data were summarized in Tables 4 and 5.

## 3. Results

### 3.1. Case Series of 15 Patients

#### 3.1.1. Case 1—BLAISE—Unilateral—Left Upper Extremity

A 38-year-old woman with a longstanding history of asthma and seasonal allergies received the first COVID-19 vaccination (Pfizer, Inc., New York City, NY, USA) on 15 January and the second on 9 February, both at the left deltoid. Approximately 2 weeks after the second vaccination, extremely pruritic lesions began to appear approximately 10 cm distal to the vaccination site on the left upper arm. Over the next 4 weeks, papules erupted more and more distally on the left arm, the exclusive site of involvement.

Examination on 8 April showed a band of papules and plaques on the left arm that measured 1.5–3.0 cm in width and followed Blaschko’s lines. It originated 2 cm proximal to the antecubital fossa and extended distally along the dorsal aspect of the left forearm to the hand (Figure 4A). The papules, pink to violaceous in color, measured 2–3 mm in diameter and coalesced in areas to form plaques. Some of the smaller papules resembled lichen nitidus. The clinical impression was linear lichen planus versus lichen striatus.

Histopathologic examination (Figure 5A,B) revealed a poorly defined lichenoid reaction with effacement of the dermo-epidermal junction. A patchy perivascular and interstitial infiltrate extended to mid dermis. The infiltrate extended diffusely through the epidermis which showed necrotic keratinocytes throughout with focal parakeratosis and spongiosis.

#### 3.1.2. Case 2—BLAISE—Bilateral—Lower Extremities

A 54-year-old woman had a history of intraoral LP dating back to 2002. On 24 March 2021, the patient received her first vaccination in the left deltoid. She experienced a burning sensation going down her left arm for the next 3 days. Eight days after the first vaccine (V1), the patient experienced a severe burning sensation in the dorsum of her left foot. A papular eruption then appeared at that site and extended proximally in a linear fashion over the next 2 weeks on the anterior aspect to the mid leg. One day after the burning started in the left foot, the patient began to experience burning in the right foot. A similar eruption progressed proximally up the anterior aspect of the right leg (Figure 4B).

Two weeks after the second vaccine (V2), (1 July 2021), the patient experienced severe burning at the left and right popliteal fossae. Shortly afterwards, over a period of 2 weeks, papular lesions appeared and extended posteriorly and distally in a linear fashion to the mid legs. Over the next week, new lesions extended distally to the heel of the left foot. A punch biopsy was taken from the left leg. Histopathologic examination revealed a lichenoid dermatitis with several atypical features. Granuloma formation (Figure 6) and rare eosinophils were seen in the superficial dermis, features not seen in any of the other 14 cases. Considering the clinical presentation and the histopathologic features, a diagnosis of vaccine-induced BLAISE was made.

#### 3.1.3. Case 3—BLAISE, Lichen Planus—Unilateral—Left-Sided Trunk and Thigh

A 69-year-old man had two Moderna vaccinations administered at the left deltoid. A left-sided asymptomatic eruption, following Blaschko’s lines, began 1 week after V2 (Figure 4C). Brown macules that later fused to form patches first appeared on the left side of his chest and stopped abruptly at the midline. Over a period of several weeks, new macules and patches appeared on the left side of his abdomen and on the proximal left thigh. The patient presented for dermatologic evaluation 6 months after the appearance of the first lesions when several clusters of asymptomatic raised lesions appeared. Examination revealed hyperpigmented macules and patches following Blaschko’s lines on the left side of the chest, abdomen and proximal aspect of the anterior thigh. Violaceous flat-topped papules were evident in small clusters at the new sites.

Biopsies were taken from violaceous papules on the left side of the chest and anterior left thigh. Histopathologic examination (Figure 7A,B) revealed identical pathology in each specimen and met the criteria for lichen planus.

#### 3.1.4. Case 4—BLAISE, Lichen Striatus-Like—Unilateral—Left Side of Neck

A 42-year-old Indian woman first noticed an itch at the top of her left ear 10 days after receiving her second COVID-19 vaccination. The itch was followed by the appearance of small papules that flattened over time and left a residual dark pigmentation. Over the next several weeks, she noticed brown macules and small patches appearing at the anterior and inferior aspects of the left side of her neck. This distal area of involvement gradually extended proximally up the neck over the next 4 weeks to meet and join the initially involved area on the ear.

Examination revealed a dark purple and brown pigmented band, 1.5–2.5 cm in width, extending inferomedially from the posterior aspect of the left ear down the neck (Figure 8A,B) to end abruptly at the midline (Figure 8C,D). The pigmented band was strikingly unilateral and followed the lines of Blaschko. A skin biopsy from the posterior aspect of the left ear revealed lichenoid dermatitis. The superficial dermis contained a focally patchy band-like lymphocytic infiltrate that extended deeply around adnexa to involve both hair follicles and sweat glands (Figure 9). 

#### 3.1.5. Case 5—V-ILE with Inverse Component

A 68-year-old man tested positive for COVID-19 in December 2020. The patient received his first dose of the Pfizer COVID-19 vaccine on 10 March 2021 and his second dose on 31 March. Three weeks later, he presented complaining of a mildly pruritic skin eruption. The initial lesion appeared on his left thigh 2 weeks after the second vaccination. Additional lesions subsequently appeared on his distal right leg, on the flexural aspect of both forearms and in the axillary vaults. Examination revealed violaceous and slightly erythematous polygonal papules. Lesions in the axillary vaults were macular. A punch biopsy was taken from the right axillary vault. Histopathologic features (Figure 7C) met the criteria for lichen planus. The patient received a third dose (booster) of the Pfizer BioNTech vaccine in November 2021. Several months later, scattered lichenoid lesions on the right leg appeared.

#### 3.1.6. Case 6—V-ILE with Multiple Lichen Planus-like Keratosis (LPLKs)

An 80-year-old woman presented to the dermatology office complaining of extremely pruritic skin lesions on the legs which initially appeared 4 days after her first COVID-19 vaccination. Examination revealed violaceous flat-topped papules. Some of the lesions appeared to be excoriated. A diagnosis of lichen planus was suspected. Biopsies were performed from the distal left thigh and the proximal right leg. Histopathologic examination of each specimen revealed features of a bandlike infiltrate in the dermis. There was minimal hydropic change but varying degrees of acanthosis and in one biopsy, papillomatosis. Six weeks later, the patient returned for follow-up and complained of an eruption of multiple new lesions on her legs that were extremely pruritic. Multiple new lichenoid papules were observed. Biopsy of a pretibial and an ankle papule revealed lichenoid dermatitis with atypical features (Figure 5C,D). In this case, the patient was elderly and the lichenoid reaction, in part, was superimposed on some of her pre-existing keratoses.

#### 3.1.7. Case 7—V-ILE, Extensive

An 81-year-old gentleman received two Moderna vaccinations in March 2021. He began to experience moderate pruritus in the axillae and groin approximately 3 weeks after the second vaccination. A papular eruption then appeared and involved the dorsa of the hands, wrists and quickly spread to the back, buttocks and thighs. A skin biopsy showed typical features of LP at the end of June. The eruption became more generalized and the itching more severe. The patient was then hospitalized and treated with intravenous Solumedrol for 3 days at the beginning of July. This was followed by an 18-day course of oral prednisone, which was complicated by the development of herpes zoster and peptic ulcer disease.

The patient presented for dermatologic evaluation on 2 September and reported generalized itching. Examination revealed characteristic lesions of LP in a generalized distribution, but most severely involved at the upper and lower extremities (Figure 10A). Histopathologic examination of skin biopsies from the affected sites on the posterior aspect of the left thigh satisfied the criteria for lichen planus (Figure 7D). The patient was treated with oral antihistamines, narrow band ultraviolet B phototherapy sessions and etretinate 30 mg daily. After 2 weeks of therapy, he began to experience decreased pruritus and a gradual clearing of the lesions. By the end of October, the eruption had resolved.

#### 3.1.8. Case 8—V-ILE, Unilateral Left-Sided

A 62-year-old woman developed lichenoid papules on the flexural aspect of her left forearm and on the left side of her umbilicus 2–3 weeks after V2. Both vaccines were administered at the left deltoid. Violaceous lesions with Wickham striae at the left side of the umbilicus started abruptly at the midline of the abdomen and then curved superiorly at the periumbilical site (Figure 10B,C). A punch biopsy from the left forearm showed lichenoid dermatitis with atypical features of focal parakeratosis and necrotic keratinocytes at all levels of the epidermis. The lichenoid infiltrate was superficial with extension into the mid dermis.

#### 3.1.9. Case 9—V-ILE Limited to Trunk with Psoriasiform Clinical Presentation

A 65-year-old male presented for initial dermatology evaluation with a 7-month history of a persistent truncal rash. It began 10 days after V2 with severe dryness and flaking at the posterior scalp. Several days later, he noticed small scaly bumps on his entire back that were asymptomatic. The eruption quickly spread to involve the parasternal area of the chest.

Examination revealed slightly erythematous to violaceous scaly patches and papules concentrated over the central parts of the chest, superior abdomen and upper and lower back. The clinical impression was psoriasiform dermatitis versus a lichenoid eruption. A biopsy was taken from the superior lumbar area of the back and revealed lichenoid dermatitis with atypical features. Focal parakeratosis, focal decrease of the granular zone, focal spongiosis and lymphocytes and necrotic keratinocytes at all levels of the epidermis were seen. The infiltrate in the dermis was focally lichenoid and patchy with extension into the mid dermis. Correlating the clinical presentation with the histopathologic features and because drug-induced lichenoid eruptions may not be pruritic and may present with a psoriasiform clinical appearance [34]. a diagnosis of V-ILE was made.

#### 3.1.10. Case 10—V-ILE with Solitary Lesion, Forme Fruste Presentation

A 40-year-old male developed a moderately itchy violaceous papule on the left forearm 2 weeks after V2. The solitary lesion darkened over time, slowly enlarged and remained occasionally itchy. The patient presented to the Mount Sinai Hospital dermatology clinic 6 months later for an unrelated problem. Examination revealed a brown to slightly violaceous, polygonal plaque measuring 1.4 × 1.0 cm located at the lateral aspect of the left antecubital fossa. A punch biopsy from the lesion revealed histopathologic features of lichenoid dermatitis. Atypical features included focal parakeratosis and spongiosis, necrotic keratinocytes and lymphocytes at all levels of the epidermis. The infiltrate was focally lichenoid or patchy in the dermis.

#### 3.1.11. Cases 11, 12, 13—V-ILE—Typical-Appearing Papular Eruptions

Case 11: A 62-year-old male developed moderately pruritic papular eruptions on the flexural aspect of the forearms and on the lower back after receiving each of his first three doses of the Pfizer COVID-19 vaccine—that is 11 days after V1, 4 days after V2 and then again 3 weeks after his first booster vaccination. Examination revealed typical-appearing lichenoid papules on the lower back and on the mid and distal flexural forearms. Some areas showed Koebnerization (Figure 11A). Punch biopsy from a lichenoid papule on the right forearm showed lichenoid dermatitis with the presence of dermal plasmacytoid dendritic cells (Figure 11B,C).

Case 12 presented in a similar manner to case 11, but with more numerous discrete lichenoid papules on the flexural aspect of the forearms and on the pretibial area of the legs.

Case 13 presented with numerous faintly violaceous papules scattered on the upper and parasacral areas of the back. There were several violaceous shiny papules on the flexural aspects of the left forearm that showed Koebnerization.

#### 3.1.12. Case 14—V-ILE with Lichenoid Plaques on Chest and Thoracolumbar Areas

A 60-year-old woman complained of a persistent rash on her back and chest that had been present for 6 months. The rash started 1 week after receiving the Janssen vaccine. Moderately intense pruritus was followed by the development of papules which evolved into larger plaques. Examination revealed large lichenoid plaques on the right side of the chest and on the thoracolumbar area of the back. This latter location showed a vertically oriented scaly violaceous plaque at the midline which extended laterally into the paraspinal areas. Smaller hyperpigmented plaques and patches were visible at the periphery. Skin biopsy of the large plaque on the back revealed lichenoid dermatitis and pronounced orthohyperkeratosis. An interesting finding was the presence of plasma cells in the dermal infiltrate.

#### 3.1.13. Case 15—V-ILE with Extensive Post Inflammatory Hyperpigmentation

A 69-year-old Black woman complained of multiple dark lesions on her back and legs stating that the discoloration began with “pink/red” patches that were mildly pruritic and evolved over a few weeks to a dark brown color. The initial lesions “erupted” on her mid lower back and flexural forearms 4 days after the second Pfizer-BioNtech booster vaccination. She had no skin eruptions after the first three doses of the vaccine which had been all from the same manufacturer.

One week later, the patient experienced an eruption of “pink/red” papules that started on her feet and ankles. It then progressed proximally up the legs and thighs over a period of 2 weeks. The eruption on the lower extremities was accompanied by an intense burning sensation for 2–3 days that was replaced by a severe itch which persisted for several weeks, at which point, the lesions flattened and healed with a dark brown discoloration in the skin.

Physical examination revealed multiple dark brown patches measuring 1–4 cm extensively distributed over the thoracolumbar area of the back, lower extremities and to a lesser extent on the flexural forearms. A review of the biopsy specimen demonstrated evidence of a “cell-poor” lichenoid dermatitis with the following features: orthokeratosis, eosinophilic hypertrophy of keratinocytes, basal cell squamatization (flattening), focally lichenoid interface dermatitis with effacement of dermo-epidermal junction, scattered colloid bodies and melanophages in the superficial dermis.

#### 3.1.14. Summary of Clincal and Atypical Histopthologic Features of Case Series 

The clinical and atypical histopathologic features of this case series are summarized below in Table 1 and Figure 12 and Figure 13, respectively.

### 3.2. Results—Literature Review

#### COVID-19 Vaccine-Associated Lichenoid Eruptions

A review of the world literature describing COVID-19 vaccine-associated cases of cutaneous lichen planus and lichenoid eruptions is summarized in Table 2. The search produced twenty-eight articles that discussed 31 cases of COVID-19 vaccine-associated lichenoid eruptions [35,36,37,38,39,40,41,42,43,44,45,46,47,48,49,50,51,52,53,54,55,56,57,58,59,60,61,62,63].

The first reports were published in March and July of 2021 [35,36]. Hiltun et al. [35]. described a 56-year-old woman with a prior history of LP who experienced reactivation 48 h after the second dose of the Pfizer vaccine. Merhy et al. [36]. reported a de novo case of LP one week after the first dose of the Pfizer vaccine. An additional case of new onset LP was listed in a footnote in a registry-based study of 414 cases of cutaneous reactions reported after Moderna and Pfizer COVID-19 vaccination [37].

In October 2021, McMahon et al. [38]. published a study of 58 cutaneous reactions with biopsy results that had been reported to an international COVID-19 dermatology registry. Four patients were reported to have had “lichen planus-like” reactions. The pathology showed lichenoid interface dermatitis with or without dermal eosinophils. Since that time, published case reports [39,40,41,42,43,44,45,46,47,48,49,50,51,52,53,54,55,56,57,58,59,60,61,62] and one small case series [63] cumulatively report on 29 additional patients. New-onset and reactivation of oral lichen planus has also been reported [43,64,65].

### 3.3. Data from Literature Review Compared to That of from Case Series

The data from the literature review were extracted and compared side by side to those of the case series, as shown in Table 3 and Figure 14, Figure 15, Figure 16, Figure 17 and Figure 18.

#### 3.3.1. The COVID-19 Vaccines

Fourteen of the fifteen patients in this case series received the Pfizer BionTech (n = 8) or Moderna (n = 6) mRNA vaccines. The remaining patient received the Janssen vector-type vaccine. This data reflects the COVID-19 vaccines that were first approved and made available in the United States at the beginning of the vaccination campaign. By contrast, patients that were studied in case reports from the literature review were located around the world. Inactivated and vector-type vaccines comprised a higher percentage of vaccines administered to that group (Figure 13).

#### 3.3.2. V-ILE – First Eruption and Vaccine Dose Number

Seventeen of the 31 patients (55%) from the literature review group experienced their first eruption after the first dose of the vaccine, as opposed to 4 of 15 patients (27%) from the case series (Figure 12). This may, in part, be explained by the fact that a far larger percentage of the literature review group (26%) received the Janssen vaccine, which is one dose, as opposed to the two-dose protocol of the Pfizer and Moderna vaccines. By contrast, only one patient from the case series group (7%) received the Janssen vaccine.

#### 3.3.3. Time to Onset of First Eruption

For the case series group, the mean time to onset after the first vaccine was 7.5 days (4 cases), while it was 14.2 days after the second vaccine (10 cases) (Figure 14). For the literature review group, the mean time to onset after the first vaccine was less than one day longer at 8.5 days. However, it was significantly shorter (7.9 days) for those cases whose first eruption occurred after the second vaccine.

#### 3.3.4. Treatment

A large proportion of eruptions in the both the case series (73%) and in the literature review (58%) were resolved with the application of topical ointments (Figure 17). Three patients (20%) in the case series and eight cases (26%) in the literature review required systemic treatment to treat their eruptions.

#### 3.3.5. Clinical Course

The 15 initial eruptions in this case series were all resolved with treatment, although 9 (60%) were resolved with hyperpigmentation (Figure 18). The clinical course of twenty-one patients from the literature review was reported. Only three of the 21 eruptions (8.6%) were reported to have healed with hyperpigmentation. These disparate results between the case report and the literature review groups may have been related to the patients’ Fitzpatrick skin types, as darker skin types tend to be more likely to heal with hyperpigmentation.

Three cases (20%), cases 2, 5 and 11, reactivated after a vaccine rechallenge. An additional three cases (20%) received a subsequent dose of the vaccine (booster dose) and reported no cutaneous reaction at all. Six of the patients (19.4%) from the literature review group reactivated upon rechallenge with the next dose of the vaccine [40,41,42,53,57,58].

## 4. Discussion

### 4.1. Immunopathogenesis—Nexus between Vaccination and Autoimmune Phenomena

Conclusive links between vaccination and the development of autoimmune phenomena have been difficult to establish; this is due to several factors, including the relative rarity of the events, the timing of events, which can include a latency period between immunization and autoimmunity, and the fact that the criteria for causality are not well-established [66,67]. There are several examples where it is believed that a causal relationship may be present. One is an association between the 1976 swine influenza vaccines and the development of a form of Guillain–Barre syndrome, a conclusion based on the strength of the epidemiologic evidence [68,69]. Another example is the association of a form of rabies vaccine prepared in brain tissue with the development of encephalomyelitis [70], hypothesized to be due to the presence of central nervous system tissue within the vaccine, inducing cross-reactive antibodies and T cells [71,72].

Although the precise mechanism by which COVID-19 vaccines might trigger LP-like disease remains to be defined, though several theories have been proposed which may help explain how infection or vaccination triggers T cell-mediated autoimmunity. These include molecular mimicry, epitope spreading and bystander activation [66]. Molecular mimicry theory (MMT) has been linked to several diseases, such as multiple sclerosis, Guillain–Barre syndrome, rheumatoid arthritis and type 1 diabetes mellitus [73,74,75], and is the most widely discussed theory in the context of vaccine-induced lichenoid eruptions. Application of this theory to the case of a COVID-19 vaccine-induced lichenoid eruption would propose the hypothesis that a shared epitope exists between a foreign peptide, such as the spike protein (S), and a self-antigen in the skin of a genetically susceptible individual. These shared epitopes need not have an exact amino acid sequence homology [75]. The initiation of an autoimmune response is then favored with the resultant activation of autoreactive T cells. This, in turn, leads to the enhanced recruitment of activated T lymphocytes from the circulation into the skin. A cascade of events with the production of proinflammatory cytokines results in apoptosis of keratinocytes. Key cytokines involved in this cascade of events are gamma interferon, tumor necrosis factor-alpha and interleukins including IL-2 and IL-6 [76].

In the case of Blaschkoid eruptions, it could be postulated that clones of cells of varying genetic lineage developed in specific lines of Blaschko and that keratinocytes in these clones share epitopes with S, providing the basis for a potential cross-reactivity. Following vaccination, the epitopes in these vulnerable keratinocytes within Blaschko’s lines become unmasked, resulting in a breakdown of self-tolerance.

However, antigenic mimicry is not by itself the complete explanation for pathologic tissue cross reactivity. Mimicry might also arise as a secondary phenomenon. An alteration of host antigenic determinants may be the result of injury at the dermal-epidermal junction with resultant formation of neoepitopes. Tissue injury may also expose epitopes that were previously cryptic and this leads to the development of self-reactive T Cells. This expanded T cell response over time from the dominant epitope to neoepitopes or cryptic epitopes is known as epitope spreading [75].

Bystander activation refers to the T cell receptor-independent and cytokine-dependent activation of T cells without exogenous antigen recognition. Tissue damage releases previously sequestered antigens that activate autoreactive T cells that were uninvolved with the initial immune response. This bystander response is rapidly induced by cytokines or Toll-like receptor (TLR) agonists. The cytokines that induce bystander activation, such as type-1 interferons, generally overlap with those that regulate the activation of antigen-specific CD8+ T cells [77,78]. TLR7/8 are activated by the mRNA vaccines, while TLR9 is activated by the viral vector vaccines [76]. Future studies may focus on the role of these TLRs and type-1 interferons in the bystander activation of T cells during COVID infection and following vaccination.

### 4.2. Timing of the Lichenoid Eruption in Relation to Vaccine Dose or Prior History

The cascade of events that lead to a reactivation of disease within lesions of fixed drug eruption (FDE), as outlined by Sontheimer^13^, may serve as a model for events within new onset or the reactivated lesions of a vaccine-induced lichenoid eruption. This may help explain why some cases first erupt after V1, while others first erupt after V2 or after subsequent booster doses of the vaccine. One may theorize that in the context of vaccine-induced lichenoid eruptions, it is possible, in view of the high infectivity of the COVID-19 virus, that some of the cases, especially those that reacted after V1, may have previously encountered the COVID-19 virus asymptomatically without mounting a sufficient antibody response that allowed for conversion to seropositivity. However, previous exposure to the virus may have resulted in the persistence of autoaggressive virus-specific memory T cells indigenously residing in the skin. Despite being incompetent to overcome self-tolerance and mount an autoimmune attack on a self-antigen in the skin after initial exposure to the viral antigens, these COVID-19-specific memory T cells may be available and now competent after vaccination to cross-react with the self-antigens in the skin and cause epidermal damage. It may, therefore, be hypothesized that lichenoid eruptions occurring relatively soon after the COVID-19 vaccination, especially those that occurred after V1, may in part be due to this pool of memory T cells, which when combined with the superimposed immunologic effect of the vaccine, i.e., a “second hit’, initiated an autoimmune attack on the skin at that time.

In addition to a prior subclinical exposure to the virus itself, a history of prior episode(s) of lichen planus could also result in this same availability of memory T cells. In this sense the vaccine acts as a trigger to reactivation of disease in a genetically vulnerable patient who was rendered more susceptible by a prior episode of a lichenoid eruption. The effect of the immune system on the involved skin in such a patient following vaccination involves crossing the necessary threshold to go from a state of self-tolerance to autoimmunity. This relationship between prior exposure to the virus or history of a lichenoid eruption and subsequent COVID-19 vaccination could then be a possible explanation for the variability in latent periods seen in the populations of patients from the literature review and present case series. The persistence of autoaggressive memory T cells in the skin could also explain those cases, in which reactivation was observed after consecutive COVID-19 vaccinations. The magnitude of the effect of these cells may not only determine whether an autoimmune attack on the skin is initiated, but also may determine the latency period after which the attack occurs and the severity of the clinical eruption.

### 4.3. Etiology

While the etiology of LP has yet to be fully elucidated, LEs can resemble LP and may often be indistinguishable from LP, clinically or histologically. LEs may be triggered by topical agents, including the color photography developer paraphenylenediamine [79] and dental restorative materials [80]. Systemic agents associated with LP and lichenoid tissue reactions include medications, viral infections and vaccines [23,24,25,26,27,28,29,30,31,32,33]. The list of medications that can cause lichenoid drug eruptions is long and growing steadily. Numerous viral infections have been implicated, including hepatitis C [24,25], hepatitis B [26], varicella-zoster [27], human herpesvirus type 7 [28], AIDS [29], Epstein–Barr virus^30^ and most recently, COVID-19 [31].

LS has also been observed to be triggered by viral infection [81,82]. One case of LS after scarlet fever was reported [83]. Jones et al. reviewed the literature of LS after vaccination (five cases) and reported an additional patient whose eruption occurred after hepatitis B vaccination [84]. Most recently, a small case series of four children with lichen striatus following COVID-19 infection was reported [85].

Vaccination-associated LP and LEs are well-established in the literature. A 2017 review of LP and lichenoid drug eruption (LDE) cases reported to the Vaccine Adverse Event Reporting System national database in the United States from July 1990 to December 2014 found that of the 33 vaccine-associated cases the vaccines with the greatest number of reports were hepatitis B (n = 8), influenza (n = 6) and herpes zoster (n = 5) [32].Vaccinations with more sporadic associations included combination hepatitis A and B, anthrax, tetanus-diphtheria-acellular pertussis (Tdap), hepatitis A, rabies, streptococcus pneumoniae and varicella. The median time to onset in this study was 14 days. The clinical and histopathologic features of idiopathic lichen planus (ILP), LDE and V-ILE are compared in Table 4.

### 4.4. Discussion of V-ILE, BLAISE: Cases 1,2,3,4

It is intriguing that four of the fifteen patients with vaccine-induced cutaneous eruptions in this case series (27%) presented with a rare distribution in the skin and were classified within the spectrum of eruptions termed BLAISE. These cases were distributed in the lines of Blaschko, an invisible series of lines on the skin that represent migration pathways of epidermal cells during embryogenesis. These lines represent a classic pattern of mosaicism, two or more genetically different cutaneous cell populations derived from a single zygote.

While the exact cause of a lichenoid eruption appearing in a Blaschkoid distribution is presently unknown, it is not due to the Koebner or isomorphic phenomena. The distribution is also not properly referred to as “zosteriform”, since it does not follow a dermatomal distribution. The term “linear lichen planus” has been used inconsistently, as it includes cases of Blaschkoid LP and BLAISE, as well as cases due to koebnerization and the isomorphic phenomenon.

The estimated incidence of the Blaschkoid presentation of LP in the literature has been based on “linear lichen planus”, which is estimated to be present in only 0.24–0.62% of patients with LP [86]. By contrast, 3 of the 31 patients in this literature review presented with a Blaschkoid distribution. Adding the four patients in our case series yields a total of seven patients with a denominator of 46 (15.2%). It, therefore, seems plausible that a causal link exists between COVID-19 mRNA vaccination and the Blaschkoid lichenoid eruptions. Nevertheless, the manifestation of lichenoid eruptions in a Blaschkoid distribution may be multifactorial in its etiology and likely includes an underlying genetic predisposition.

### 4.5. Variability of the Blaschkoid Clinical Presentation: Symptoms, Distribution, Laterality, Treatment and Resolution

The symptoms in our four cases were variable ranging from no symptomatology (case 3) to severe pruritus at the time of the eruption (case 1) to a continuous burning sensation at the site of origin accompanied by pruritus at the site of the eruption (case 2). The distribution of these V-ILEs involved the extremities (cases 1,2,3), trunk (case 3) and neck (case 4). In three of the four cases, the eruption was unilateral and localized to the side where the vaccines were administered, while in one patient (case 2), the eruption was bilateral. The treatment ranged from topical triamcinolone and antihistamines to a tapering course of oral prednisone (case 2). Resolution occurred with post-inflammatory hyperpigmentation in three of the four cases.

The literature summarizing cases of Blaschkoid lichenoid eruptions occurring after vaccination is summarized in Table 5. Three reports describe four patients whose eruptions occurred following influenza immunizations [88,89,90].Three recent case reports describe unilateral lichenoid eruptions in a Blaschkoid distribution after intramuscular COVID-19 vaccination into the ipsilateral deltoid [60,61,62].These three eruptions, termed “lichen striatus,” [60] “linear lichen planus” [61] and “Blaschkolinear acquired inflammatory skin eruption” [62] by the authors, may be nosologically unified by the term BLAISE. This report describes herein an additional four cases. Of interest, one case of Blaschkoid pityriasis rosea after COVID-19 vaccination was also recently reported [91].

The appearance of the eruption along Blaschko’s lines on the left forearm of case #1 as the exclusive cutaneous site of involvement 2 weeks after receiving the vaccination in the left deltoid encourages a presumption of causality. The unilaterality of three of the four Blaschkoid cases is not unexpected as most cases of acquired Blaschkoid eruptions are unilateral. However, the exclusive localization of the lesions to the same (left) side as the vaccination injection sites in these patients strengthens the argument for a causal relationship between the vaccination and the cutaneous eruption.

The bilaterality of Blaschkoid LP is an exceedingly rare phenomenon. To our knowledge, only three such patients have previously been reported in the literature [94,95,96]. Case 2 in the series reported herein represents the fourth patient with a lichenoid eruption following the lines of Blaschko bilaterally. The complicated history of this patient’s V-ILE with a chronological sequence of linear eruptions on her legs of anterior-posterior-anterior strongly suggests that it was caused by the vaccinations. It may be suggested that her prior history of LP, although it was 19 years prior to the first COVID-19 vaccination, triggered dormant autoimmune disease to reactivate after the first dose of the vaccine. Additionally, this patient’s flare after the second vaccination could be said to represent a “positive rechallenge”.

## 5. Conclusions

A systematic review of the clinical and histopathological features of COVID-19 vaccine-induced lichenoid eruptions as distilled from the prior literature and investigated in this case series supports the conclusion that a spectrum of features exists for each. The clinical variability within this spectrum has been discussed above and is summarized in Table 1. The spectrum of histopathologic features within the case series is discussed below.

### Variability of Pathology within the V-ILE Case Series

Lichenoid tissue reactions can show a polymorphous histology and these 15 cases were no exception. The initial classification of these cases divided them into two groups based on whether their histologic features met the criteria for stereotypical lichen planus (Figure 19). Eleven cases, including three with a Blaschkoid distribution (cases 1,2 and 4) formed the first group. This group showed some of the stereotypical changes of lichen planus, i.e., a bandlike dermal infiltrate and vacuolar changes at the dermoepidermal junction associated with rare necrotic keratinocytes. However, the criteria for the diagnosis of true lichen planus were incompletely satisfied.

Nine of these 11 cases formed group 1b, which additionally showed atypical changes, i.e., more extensive necrosis of keratinocytes, focal parakeratosis and spongiosis (Figure 5, Figure 12 and Figure 13). This latter change is often seen in lichenoid reactions from an exogenous source, such as oral medications or topical agents and may, in some cases, represent an evolving tissue reaction that over time culminates into a more fully developed LR [97]. Case The atypical findings seen in this group of cases should trigger a heightened suspicion for an exogenous agent such as a medication or vaccine.

Despite the Blaschkoid presentation in four of the fifteen cases, there was evidence of deep dermal and eccrine involvement, changes often associated with lichen striatus, in only one case.

The second group of cases (3, 5, 7 and 8) included one patient (case 3) with a Blaschkoid distribution. All cases in this group showed the typical features of lichen planus. This is not surprising as triggers for lichenoid tissue reaction can sometimes produce a histology that meets the criteria of lichen planus and does not always produce a different histology. While case 8 (group 2b) did show additional atypical features including extensive necrosis of keratinocytes at all levels of the epidermis, three cases (group 2a: cases 3, 5 and 7) mimicked the standard histology of lichen planus exactly and did not deviate from it (Figure 8).

We believe that the unique histology seen in groups 1b and 2b may have predictive value in that their atypical features offer important etiologic clues. These could facilitate a distinction from idiopathic lichen planus in the appropriate clinical context and heighten the suspicion for an exogenous agent. In the absence of a confounding history of a recent viral infection or initiation of a new medication, the history of a recent vaccination may help confirm the diagnosis of vaccine-induced lichenoid eruption.

## 6. Summary

Lichenoid eruptions, whether arising de novo or as a reactivation of a previously existing condition, should be added to the list of inflammatory skin diseases that are associated with COVID-19 vaccination. It is likely that this association is not fully appreciated and that the true number of V-ILEs linked to this vaccine has been underreported. Nevertheless, almost all patients responded well to treatment and healed completely with or without post-inflammatory hyperpigmentation. This paper represents the largest case series and describes reactions that are characterized by variable phenotypes, both clinically and histopathologically. The Blaschkoid distribution of lesions, although only a rare presentation of idiopathic lichen planus, appears to be significantly linked to the COVID-19 mRNA vaccines. Although a causal relationship between COVID-19 vaccination and LEs is difficult to establish conclusively, we believe that physicians should approach all such eruptions with a heightened index of suspicion and specifically question patients with regards to their vaccination history. The time of onset of a V-ILE after the administration of the vaccine and the distribution of the eruption may offer important clinical clues to its diagnosis. An appreciation of the variability and the scope of histopathologic features of V-ILE, especially when correlated with the clinical history and presentation, may be confirmatory. Furthermore, additional investigative studies regarding the immunopathology of this skin disease and its associated inflammatory signaling pathways may offer insight into other Th1-driven autoimmune phenomena related to COVID-19 vaccination. All authors attest that they meet the ICMJE criteria for authorship.

## Figures and Tables

**Figure 1 vaccines-11-00438-f001:**
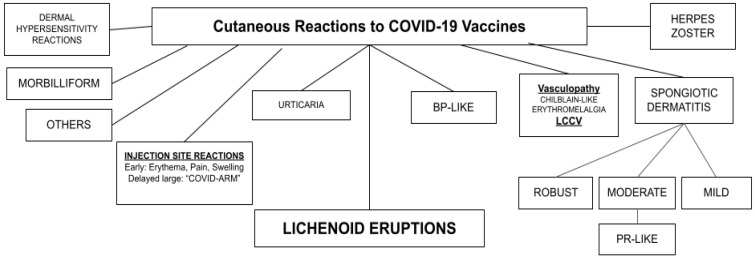
Cutaneous reactions to COVID-19 vaccines. BP, bullous pemphigoid; LCCV, leucocytoclastic vasculitis; PR, pityriasis rosea.

**Figure 2 vaccines-11-00438-f002:**
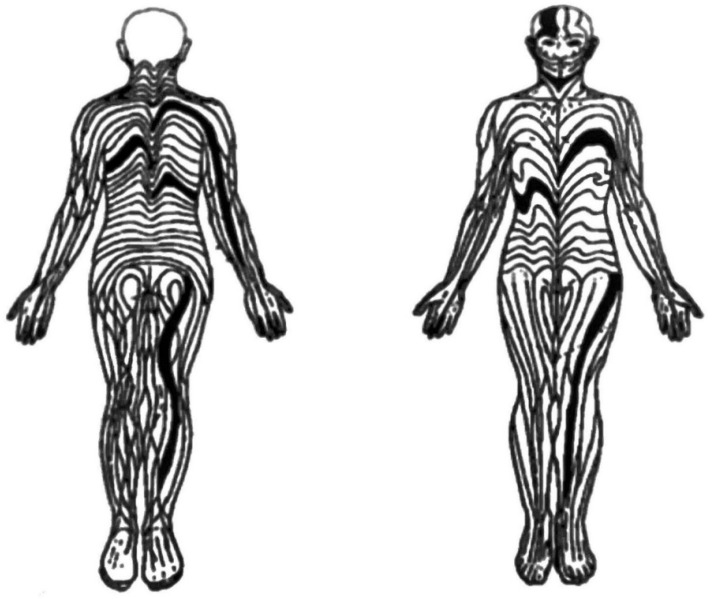
Blaschko’s lines, posterior and anterior views, as illustrated in the 1901 supplement to the Proceedings of the German Dermatological Society meeting [19].

**Figure 3 vaccines-11-00438-f003:**
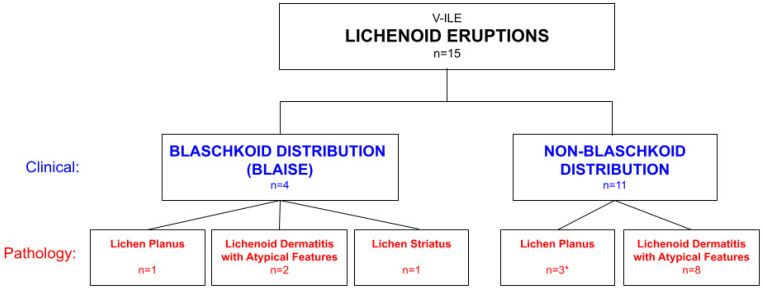
Classification of 15 patients with COVID-19 vaccine-induced lichenoid eruptions based on the clinical distribution of the eruption and pathology diagnosis.

**Figure 4 vaccines-11-00438-f004:**
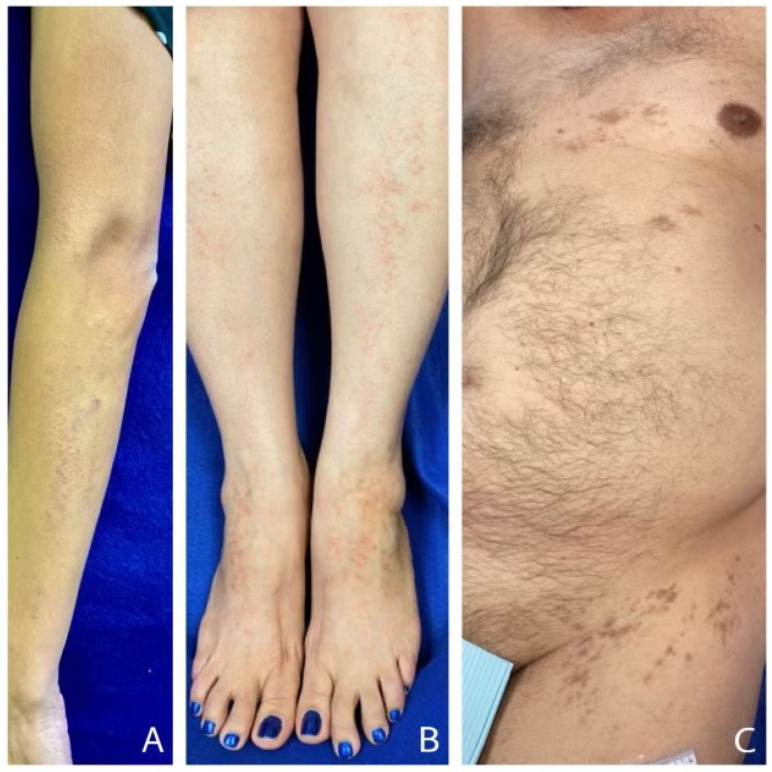
COVID-19 vaccine-induced lichenoid eruptions (V-ILEs) distributed in Blaschko’s lines. (**A**) unilaterally on the left forearm (case 1), (**B**) bilaterally on the anterior aspect of legs (case 2) and (**C**) unilaterally on the left side of chest, abdomen and proximal thigh (case 3).

**Figure 5 vaccines-11-00438-f005:**
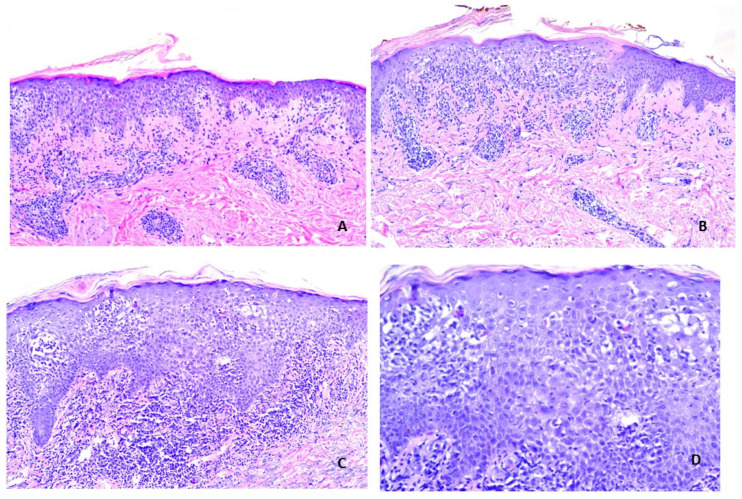
V-ILEs, Cases 1 and 6. These cases showed atypical features in an otherwise lichenoid eruption. (**A**,**B**) V-ILE Case 1. Each of two biopsies showed a poorly defined lichenoid infiltrate with spongiosis. (Hematoxylin-eosin stain; Original magnification, ×100). (**C**,**D**) V-ILE Case 6B, second biopsy. There is a poorly defined lichenoid infiltrate and necrotic keratinocytes at all levels of the epidermis associated with spongiosis. (**C**,**D**) Hematoxylin-eosin stain; Original magnification, ×100 ×200).

**Figure 6 vaccines-11-00438-f006:**
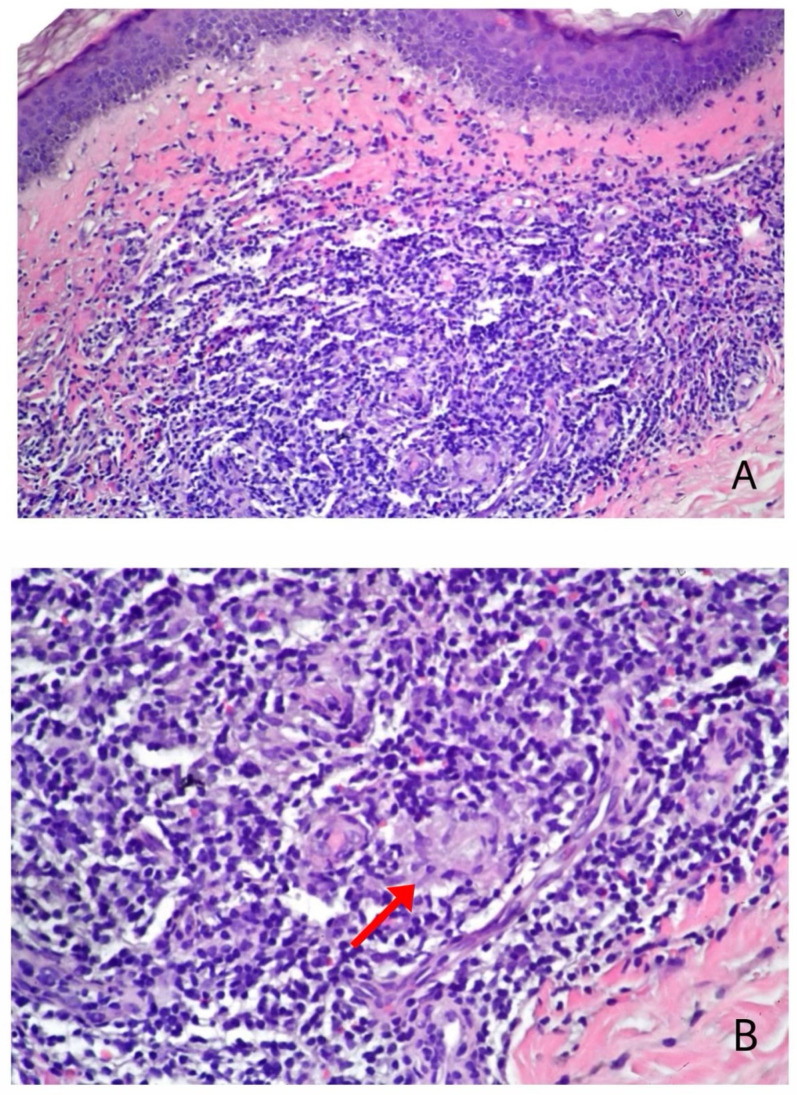
V-ILE Case 2. (**A**) The dermis shows a bandlike mononuclear infiltrate. Interspersed are scattered collections of epithelioid histiocytes. (**B**) Higher power showing epithelioid histiocytes (arrow). (**A**,**B**) Hematoxylin-eosin stain; Original magnification. (**A**), ×100; (**B**), ×200.

**Figure 7 vaccines-11-00438-f007:**
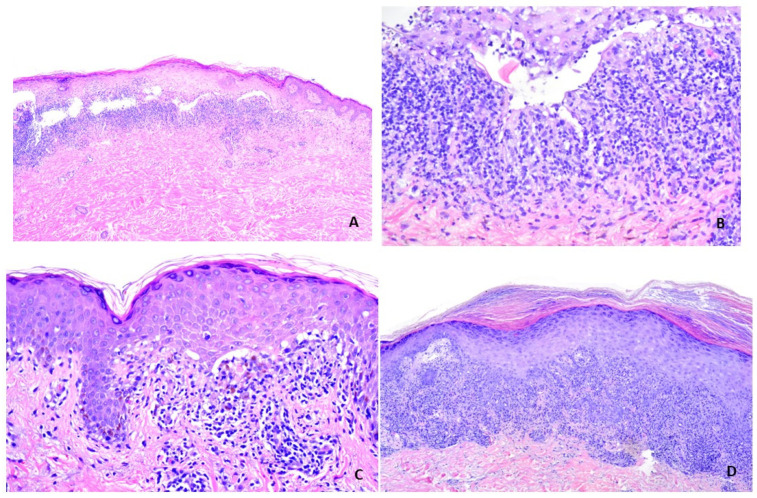
V-ILEs showing stereotypical features of LP. (**A**,**B**), Case 3. (**A**) There is a bandlike infiltrate associated with Max-Joseph spaces. (Hematoxylin-eosin stain; Original magnification, ×100). (**B**) There is a bandlike infiltrate associated with prominent Max-Joseph spaces, vacuolar change and necrotic keratinocytes. (Hematoxylin-eosin stain; Original magnification, ×100 ×200). (**C**) Case 5. There is a bandlike infiltrate associated with Max-Joseph spaces and hypergranulosis. (Hematoxylin-eosin stain; Original magnification, ×200). (**D**) Case 7. There is a bandlike infiltrate associated with wedge-shaped hypergranulosis. (Hematoxylin-eosin stain; Original magnification, ×100).

**Figure 8 vaccines-11-00438-f008:**
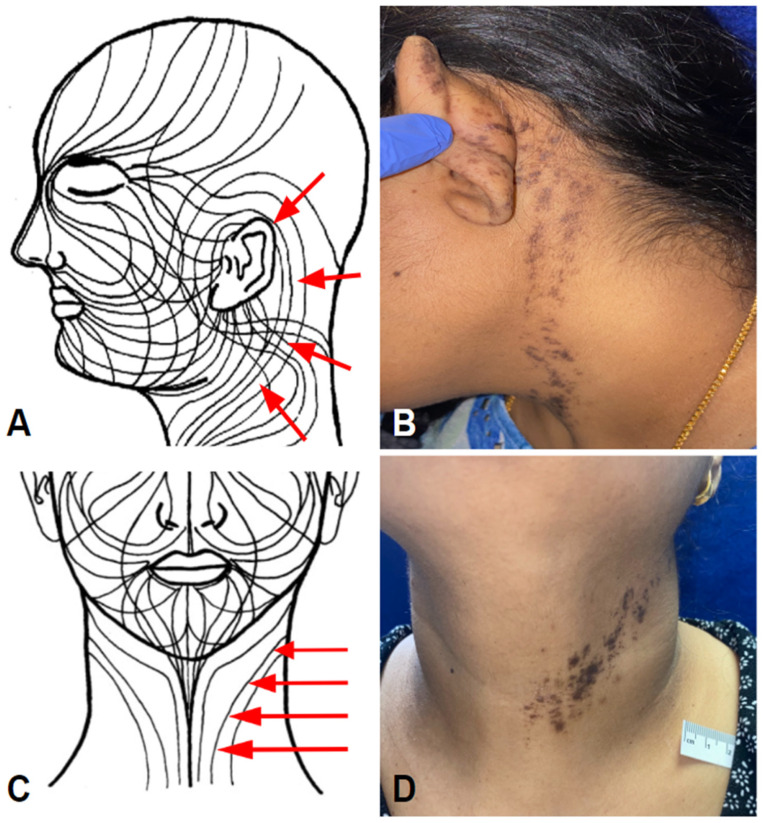
V-ILE Case 4, lichen striatus-like. Schematics showing Blaschko’s lines on head and neck; (**A**) lateral and (**C**) frontal views. Reproduced from Happle, R.; Assim, A. *J. Am. Acad. Dermatol.* **2001**, *44*, 612–615. (**B**) Starting superiorly and posteriorly at the left ear. (**D**) A purple-brown band extends inferiorly and medially to terminate abruptly at the midline of the neck.

**Figure 9 vaccines-11-00438-f009:**
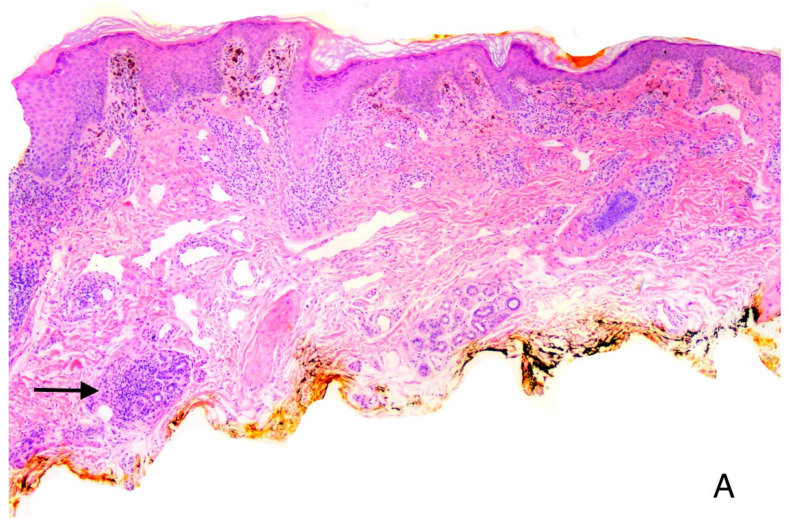
V-ILE Case 4, lichen striatus-like. (**A**) Lichenoid dermatitis, patchy in areas, with a superficial and deep lymphocytic infiltrate. The deep infiltrate surrounds eccrine glands (arrow). Prominent pigmentary incontinence is seen in the papillary dermis. (Hematoxylin-eosin stain; Original magnification, ×100). (**B**) Perieccrine lymphocytic infiltrate is shown at a higher magnification (arrow). (Hematoxylin-eosin stain; Original magnification, ×200).

**Figure 10 vaccines-11-00438-f010:**
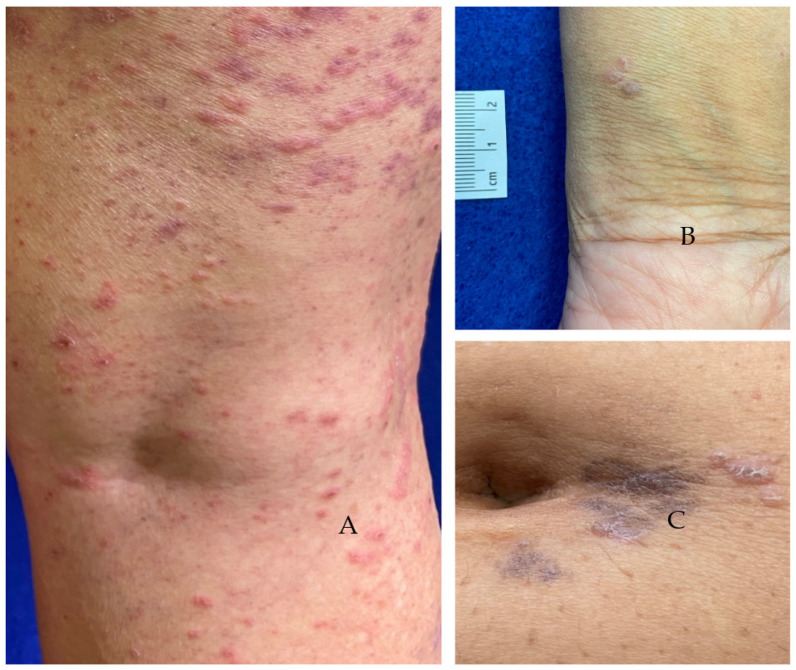
V-ILE. (**A**)**,** Case 7, 81-year-old male and (**B**,**C**) Case 8, 62-year-old female. (**A**) Numerous violaceous papules and plaques on the posterior aspect of the thigh and proximal leg. (**B**) Violaceous flat-topped papules clustered at the flexural aspect of the left wrist. (**C**) Violaceous papules and plaques in the periumbilical area of the abdomen respecting the midline. Wickham’s striae are evident.

**Figure 11 vaccines-11-00438-f011:**
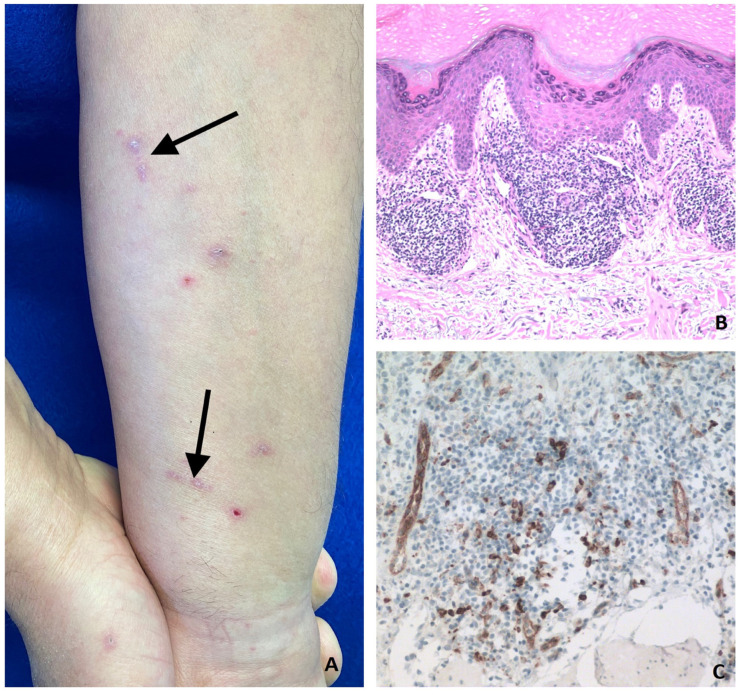
V-ILE Case 11. (**A**) Koebnerization, a linearly-distributed grouping of lesions precipitated by trauma (scratching in this case), is seen on the mid and distal flexural aspect of the left forearm (black arrows). Several excoriated papules are also shown. (**B**) Lichenoid dermatitis. Compact orthokeratosis, wedge-shaped hypergranulosis, acanthosis, foci of hydropic change and colloid bodies—features characteristic of typical lichen planus—are seen. The dermal lymphocytic infiltrate is focal and patchy with involvement of both the superficial and deep dermis. (Hematoxylin-eosin stain; Original magnification, ×100). (**C**) Anti-human CD123 mouse monoclonal antibody revealed plasmacytoid dendritic cells in the dermis. These antigen-presenting cells, seen in the dermal infiltrates of inflammatory dermatoses, such as psoriasis, cutaneous lupus and lichen planus, are not seen in normal skin. Lighter staining of endothelial cells within venules is also seen. (Anti-CD123 immunoperoxidase; Original magnification, ×400).

**Figure 12 vaccines-11-00438-f012:**
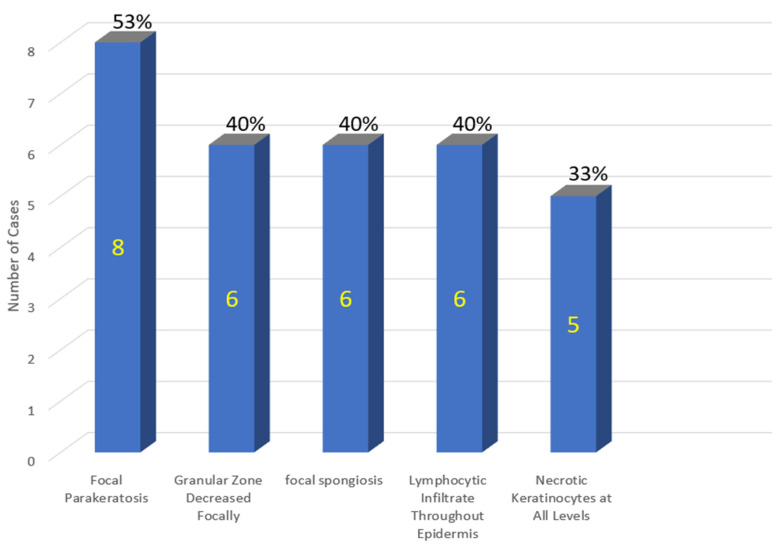
COVID-19 vaccine-induced lichenoid eruptions—case series of 15 patients. Shown on the x-axis are atypical features of the epidermis.

**Figure 13 vaccines-11-00438-f013:**
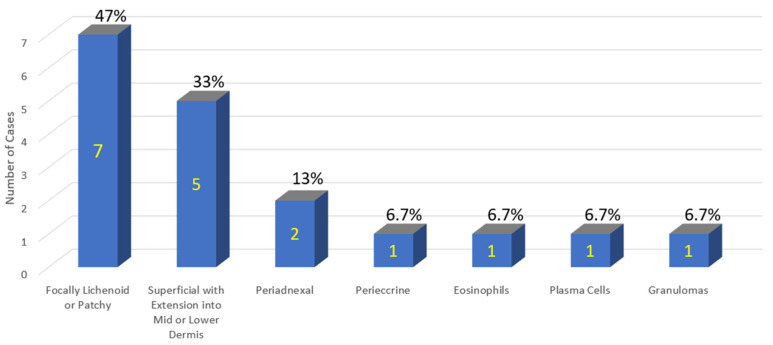
COVID-19 vaccine-induced lichenoid eruptions—case series of 15 patients. Shown on the x-axis are atypical features of the dermal infiltrate.

**Figure 14 vaccines-11-00438-f014:**
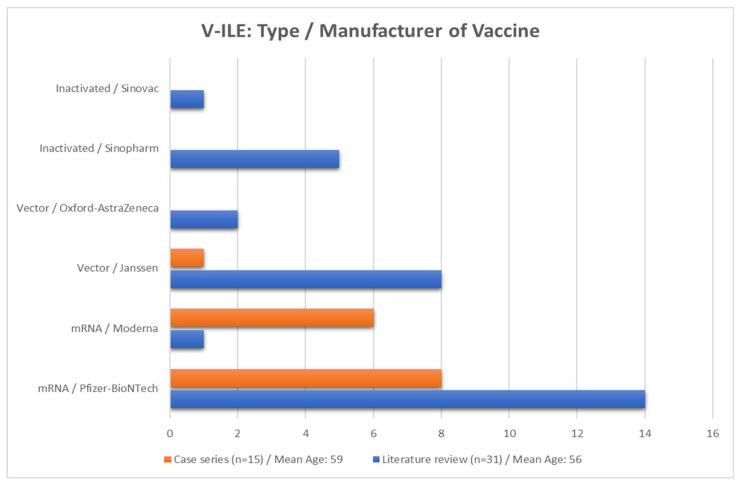
Types and manufacturers of the COVID-19 vaccines administered in the cases from the case series and review of the literature. n = number of cases.

**Figure 15 vaccines-11-00438-f015:**
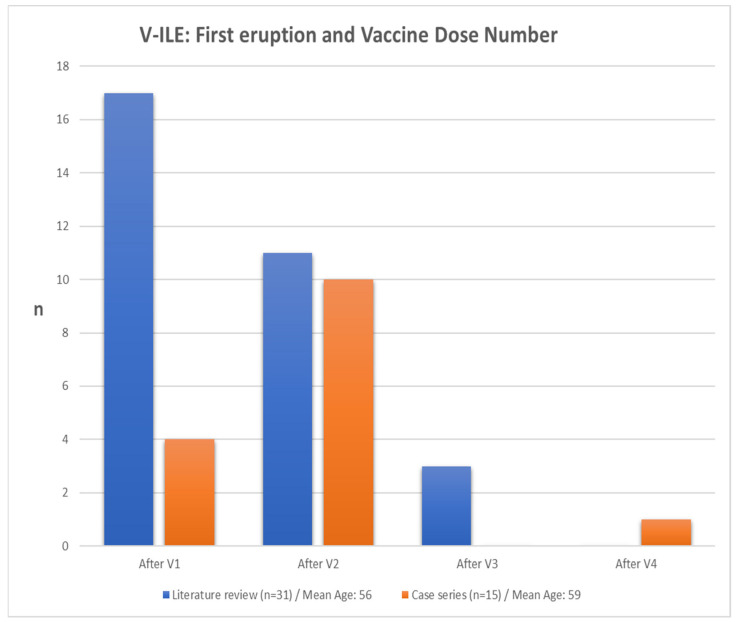
Number of cases from the case series and literature review experiencing first eruption and corresponding vaccine dose number. n, number of cases; V1, first dose of vaccine; V2, second dose of vaccine; V3, third dose of vaccine; V4, fourth dose of vaccine.

**Figure 16 vaccines-11-00438-f016:**
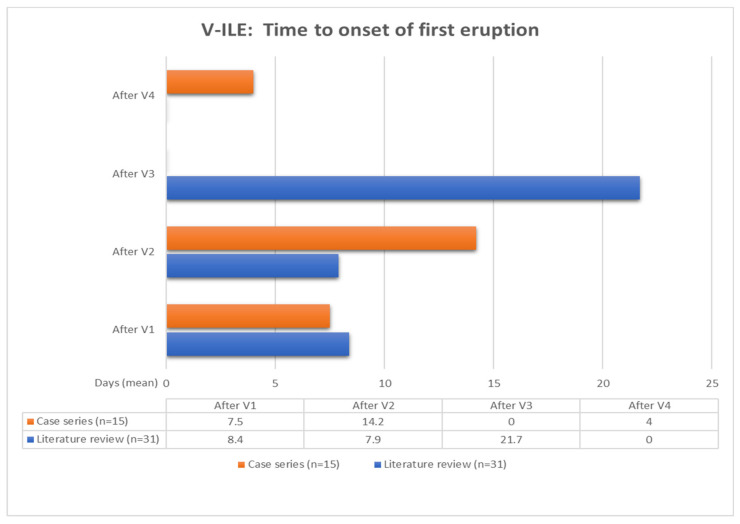
Key: n = number of patients; V1 = first COVID-19 vaccine; V2 = second dose of COVID-19 vaccine; V3 = third dose of COVID-19 vaccine; V4 = fourth dose of COVID-19 vaccine.

**Figure 17 vaccines-11-00438-f017:**
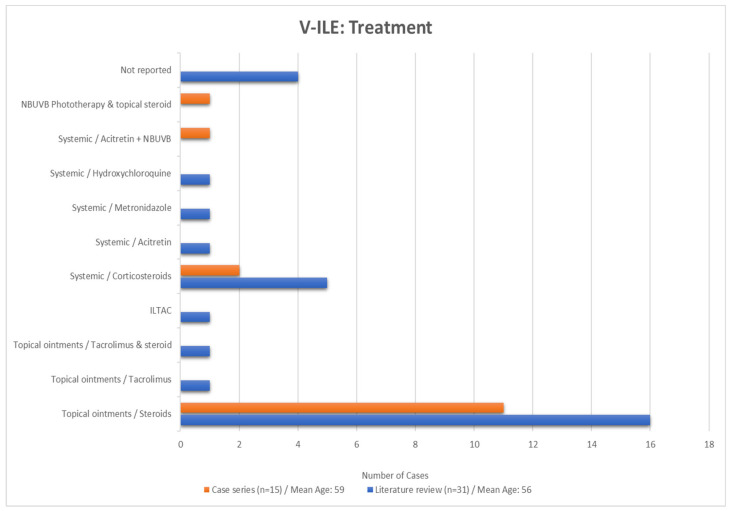
Treatments employed to treat the cases in the series and literature review groups. NBUVB, narrow-band ultraviolet light phototherapy; ILTAC, intralesional triamcinolone.

**Figure 18 vaccines-11-00438-f018:**
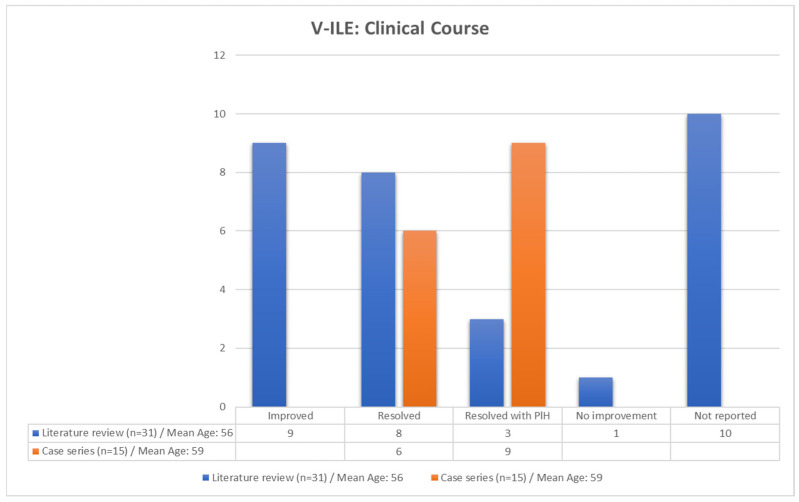
Clinical course of cases in the case series and the literature review. n = number of cases; PIH, post inflammatory hyperpigmentation.

**Figure 19 vaccines-11-00438-f019:**
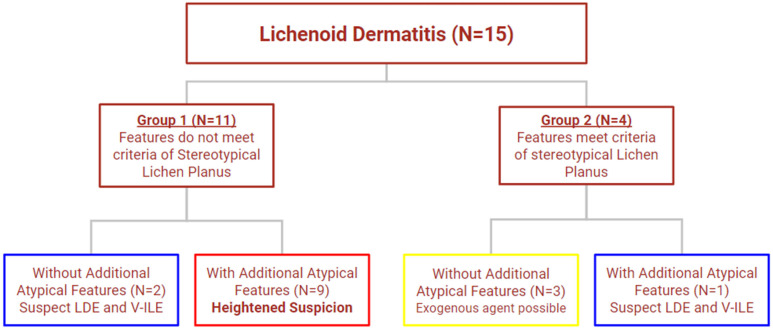
Fifteen cases of lichenoid dermatitis were classified into four groups based on whether they met criteria of stereotypical lichen planus and whether they showed additional atypical features. N, number of cases; LDE, lichenoid drug eruption; V-ILE, vaccine-induced lichenoid eruption. Key for colored boxes: Group 1a, blue; 1b, red; 2a, yellow; 2b, blue.

**Table 1 vaccines-11-00438-t001:** Vaccine-induced lichenoid eruption (V-ILE) following COVID-19 vaccination—demographics and clinical characteristics of fifteen patients from a case series.

Pt #	Age	Sex	Race	Medical History	Vaccine	Interval to 1st Eruption	Presentation	Distribution	Treatment	Outcome
1	38	F	Asian	Asthma	Pfizer	2 weeks after V2	LE-B	Unilateral/Left forearm	TAC oint, oral AH	Resolved with PIH
2	54	F	Hispanic	Asthma, oral LP	Pfizer	8 days after V1; RA 2 weeks after V2	LE-B	Bilateral/Anterior and posterior legs	TAC oint, prednisone	Resolved
3	69	M	Asian	HTN, hypothyroid, pre-DM	Moderna	1 week after V2	LE-B: LP	Unilateral/Left chest, abdomen, back, groin	TAC oint	Resolved with PIH
4	42	F	Asian	None	Pfizer	10 days after V2	LE-B: LS-like	Unilateral/Left post-auricular and neck	TAC oint	Resolved with PIH
5	68	M	Hispanic	HTN, COVID-19	Pfizer	2 weeks after V2; RA months after B1	LP with inverse lesion	Bilateral/Axilla, forearms, thigh	TAC oint, oral AH	Resolved with PIH
6	80	F	Caucasian	Anxiety	Pfizer	4 days after V1	LP with LPLKs	Bilateral/Lower extremities and forearms	TAC oint, NBUVB	Resolving
7	81	M	Asian	Prostate CA	Moderna	3 weeks after V2	Extensive LP	Bilateral/Trunk and extremities	Oral AH, NBUVB, acitretin	Resolved
8	62	F	Asian	None	Moderna	2–3 weeks after V2	LP	Unilateral/Left forearm, periumbilical	TAC oint	Resolved with PIH
9	65	M	Caucasian	GIST	Pfizer	10 days after V2	Psoriasiform	Bilateral/Trunk	TAC oint	Resolved
10	40	M	Hispanic	None	Moderna	2 weeks after V2	Solitary lichenoid lesion	Unilateral/Left ACF	TAC oint	Resolved with PIH
11	62	M	Caucasian	HTN, stroke	Pfizer	11 days after V1; RA 4 days after V2 and 3 weeks after B1	Papular LP	Bilateral/Flexural forearms and lower back	TAC oint, prednisone	Resolved
12	66	M	Asian	HTN, T2DM, HLD	Moderna	3 weeks after V2	Papular LP	Bilateral/Flexural forearms and pretibial legs	Clobetastol oint	Resolved
13	36	F	Hispanic	LP	Pfizer	12 days after V2	Papular LP	Bilateral/Upper and lower back, flexural wrists	Clobetastol oint	Resolved
14	59	F	Black	Breast CA lumpectomy	Janssen	1 weeks after V1	Truncal plaques	Bilateral R chest lower back	Clobetastol oint	Resolved with PIH
15	68	F	Black	HTN, hypothyroid	Moderna	4 days after B2	Extensive PIH	Bilateral/Forearms, back, legs	Clobetastol oint	Resolved with PIH

LE-B, lichenoid eruption-Blaschkoid; LP, lichen planus; Pre-DM, pre-diabetes mellitus; LS-Like, lichen striatus-like; LPLK, lichen planus-like keratosis; F, female; M, male; HTN, hypertension; CA, carcinoma; GIST, gastrointestinal stromal tumor; Hep C, hepatitis C; V1, first dose of vaccination; V2, second dose of vaccination; B1, first dose of booster; B2. second dose of booster; ACF, antecubital fossa; RA, reactivation; TAC oint, triamcinolone ointment; AH, antihistamine; NBUVB, narrow-band ultraviolet B; PIH, post-inflammatory hyperpigmentation; T2DM, Type-2 diabetes mellitus; HLD, hyperlipidemia; R, right side.

**Table 2 vaccines-11-00438-t002:** Lichenoid euptions following COVID-19 Vaccination—Review of the Literature.

Authors [REF #]	A G	Type of Vaccine/MFR	Type of Eruption	Time to Onset of Eruption	Treatment, Clinical Course, Distribution
Hiltun [35] 2021	56 F	mRNA/P-B	Reactivation of lichen planus	2 days after V2	Topical steroids; NR ankles, flexural wrist, forearms, periumbilical, mammary and axillary folds
Mehry [36] 2021	56 F	mRNA/P-B	New onset LP	1 week after V1	NR;NR Trunk
McMahon [38] 2021 V-REPP	NR	mRNA P-B n = 3 mRNA Moderna n = 1	Lichen planus-like NOTE: 4/58 cases with biopsy reports	NR	NR:NR NR
Herzum [39] 2021	59 F	mRNA/P-B	Flare of LP—papular	2 weeks after V2	Topical steroids; Resolved after 3 weeks. Ankles, feet
Piccolo [40]	64 F	mRNA/P-B	LP on areas of vitiligo	5 days after V1 1 day after V2	Topical and systemic corticosteroids/NR Dorsa of hands—Bilat
Bularca [41] 2022	29 F	mRNA/P-B	After V1: LP on areas of vitiligo. After V2 LP extended to areas not affected by vitiligo.	1 week after V1 LP after V2	Methotrexate Clinical course: NR Dorsa of hands, wrists, eyelids, submammary region, legs. Oral mucosa
Diab [42]	60 F 55 F	Vector/AstraZeneca Sinopharm	(1) Flare of lichen planopilaris and new lesions of LP (2) Flare of LP— previously had only a solitary lesion	(1) 2 weeks after V2 (2) 3 days after V1, more severe after V2	ILTAC, Tofacitinib Gradual improvement Face, scalp Metronidazole 500 mg bid Lesions improved Lower extremities and buttocks
Paolino [43] 2022	63 F	mRNA P/B	Palmoplantar	3 days after V2	Acitretin 25mg/day × 2 months and topical calcipotriene/betamethasone foam After 4 weeks: Total clearing of acral lesions, residual hyperpigmentation of palms. Palms, wrists and soles
Correia [44] 2021	66 M	Vector Oxford-AstraZeneca	Exuberant Generalized	5 days after V1	Topical steroid; Resolved after 4 months. Back, scalp, trunk, extremities
Onn [45] 2021	53 F	mRNA/P-B	Generalized Lichenoid skin reaction and SIRS	12 days after V1	Topical steroid, cetirizine; oral prednisone then IV hydrocortisone Abdomen, chest, back, scalp
Ziraldo [46] 2021	66 F	Vector/AstraZeneca	Lichenoid exanthema— EM-like lesions	3 weeks after V1	Oral steroids Resolved in 10 days. Entire body involved
Babazadeh [47] 2021	52 F	Sinopharm	New onset LP Clinical DX—NO BX	1 week after V2	Topical steroids, Antihistamines; Favorable response Arms, legs
Zagaria [48] 2022	54 M	mRNA/P-B	New onset LP	10 days after V1	Oral prednisone, 25 mg daily with rapid taper over 4 weeks. Rapid resolution. Trunk, arms, legs
Camela [49] 2021	59 M	mRNA/P-B	Lichenoid eruption	2 weeks after V1	NR;NR Trunk, extremities
Awada [50] 2022	44 M	Vector Oxford-AstraZeneca	Inverse LP	2 weeks after V2	Betamethasone cream Resolved after 4 weeks Axillae
Satilmis [51] 2022	60 F	Inactivated virus CoronaVac	Lichen planus	6 days after V1	Treatment: NR Flexural wrists, dorsa of hands and feet
Alrawashdeh [52] 2022	46 M	Vector Oxford-AstraZeneca	Lichen planus	5 days after V1	Prednisone was refused. Hydroxychloroquine 200 mg bid, minimal improvement after 2 months. Face, abdomen, back and legs
Sun [53] 2022	64 F	Vector Oxford-AstraZeneca	Lichen planus pigmentosus- inversus	2 weeks after V1	Topical steroid; Minor improvement after 2 months Inframammary, axillae, lower back, groin
Kurosaki [54] 2022	54 F	mRNA Pfizer	Lichen planus pemphigoides	1 day after V2	Topical steroid Blisters appeared Trunk and xtremities
Picone [55] 2022	81 M	mRNA Moderna	New-onset LP with oral LP	7 days after V1	Topical steroid and cetirizine oral AH Exam after 25 days: Resolved Flexural wrists, lower back, posterior thighs, dorsa of feet
Zengarini [56] 2022	49 M	Vector AstraZeneca	Eruptive LP	11 days after V2	Topical steroids, Oral AH Resolved with no itch and only mild erythema after 1 month. Trunk and extremities
Masseran [57] 2022	65 F	Vector ChAdOx1 nCoV-19	Extensive LP	10 days after V1 7 days after V2 which were 3 months apart.	Clobetasol cream x 4 weeks—nearly complete remission but remains with diffuse hyperpigmentation. Arms, legs, buttocks, abdomen
Gamonal [58] 2022	86 M	Vector ChAdOx1 nCoV-19	Eruptive LP	7 days after V1 Worse after V2 which was 3 months after V1.	Topical steroid cream Clinical course: NR Arms, legs, trunk, buttocks
Alotaibi [59] 2022	57 F	mRNA/Pfizer	LP	3 weeks after third dose	Tacrolimus ointment, steroid ointment Clinical course: NR Chest, axillae, arms, legs
Belina [60] 2021	42 F	mRNA P-B	Lichen Striatus	3 days after V2	Tacrolimus ointment 0.1% Result: NR
Kato [61] 2022	57 F	mRNA P/B	Linear lichen planus	2 weeks after V1 (3rd dose of the vaccine)	Topical steroids Improvement with only mild residual hyperpigmentation
Rovira-Lopez [62] 2022	46 F	mRNA/P-B	BLAISE	A few days after V1. No flare after V2.	Topical steroids No improvement
Hali [63] 2022	67 M	Vector/Astrazeneca	Lichenoid eruption	3 days after V2	Topical steroid Improved thigh, neck, upper chest, forearms
	20 F	Inactivated virus Sinopharm	Lichenoid Eeuption	1 day after V1	Topical steroid; Started to heal. Entire body
	28 M	Inactivated virus Sinopharm	Lichenoid eruption	15 days after V2	Topical steroid and antihistamines Some improvement legs, arms
	65 F	Inactivated virus Sinopharm	Lichenoid eruption	30 days after V3	Topical steroids Some improvement entire body

MFR. Manufacturer; LP, lichen planus; P-B, Pfizer BioNTech; V1, first vaccination dose; V2, second vaccination dose; V3, third vaccination dose; NR, not reported; ILTAC, intralesional triamcinolone; SIRS, systemic inflammatory response syndrome; IV, intravenous; EM, erythema multiforme; DX, diagnosis; BX, biopsy.

**Table 3 vaccines-11-00438-t003:** Summary of data from literature review and case series of cutaneous lichenoid eruptions associated with COVID-19 vaccination and confirmed by histopathology.

	Literature Review (n = 31)	Case Series (n = 15)
**Mean age (Range)** Median age (Range)	56 [20–86]	59 [36–81]
**Gender** F, n (%)	21 (68)	8 (53)
**Race, n (%)** Asian Hispanic Caucasian Black	N/A N/A N/A N/A	6 (40) 4 (27) 3 (20) 2 (13)
**Blaschkoid distribution, n (%)**	03 (10)	04 (27)
**First eruption, n (%)** After V1 After V2 After V3 After V4	17 (55) 11 (35) 03 (10) None	04 (27) 10 (67) None 01 (07)
**Time to onset of eruption** (mean days; range) After V1 After V2 After V3 After V4	8.4 (1–21) 7.9 (1–15) 21.7 (14–30) N/A	7.5 (4–11) 14.2 (7–21) N/A 4.0 (N/A)
**Eruptions after successive vaccinations**: n (%)	06 (19)	02 (13)

F, female; n, number of cases; N/A, not applicable.

**Table 4 vaccines-11-00438-t004:** Comparison of Clinical and Histopathologic Features of Idiopathic Lichen Planus (ILP), Lichenoid Drug Eruption (LDE) and Vaccine-Induced Lichenoid Eruption (V-ILE).

	Idiopathic Lichen Planus (ILP)	Lichenoid Drug Eruption (LDE)	Vaccine-Induced Lichenoid Eruption (V-ILE)
Clinical			
Mean age	Fifth or sixth decade	Sixty-six years [23]	Sixth decade
Latent Period	N/A	Often several months or longer	Several days to several weeks in most published reports.
Pruritus, Burning	Intense pruritus, common	Pruritus may or may not be present.	Intensity varies among cases. Often intense but may be completely absent in some cases. Burning sensation is possible.
Location	Flexural wrists and forearms, presacral, shins, ankles, intraoral, genitals, hair, nails	More generalized. Less likely to involve hair, nails. Intraoral, in some cases	May or may not involve classic sites like flexural wrists. Peripheral distribution, often. May be generalized.
Mucous Membranes	Commonly involved	May be involved	May be involved
Photodistribution	Not characteristic	Frequent, depending on the drug.	Not characteristic
Blaschkoid Distribution	0.24–0.62% of all cases [86]	Has been reported [87]	Several case reports [60,61,62,88,89,90,91] Cases 1–4 herein. Likely more common in V-ILE than in ILP. Present in 04/15 cases reported herein.
Morphology	Classic is “6 Ps”: Pruritic purple, planar, polygonal, papules and plaques	May show classic morphology. May show larger plaques. May appear psoriasiform or eczematous.	May show classic morphology. Papules may be skin-colored or erythematous. May be psoriasiform.
Wickham Striae	Classic	Often absent	Present in some cases [41,57,58,92]
Hyperpigmentation	Common	Very common	Very common
Number of Lesions	Most often multiple; may be solitary in lichen planus-like keratosis	Most often multiple	Most often multiple, but may be solitary (case 9)
Histopathologic			
Lichenoid Interface Dermatitis	Present	Present	Present
Compact Orthokeratosis	Characteristic	May be present	May be present
Wedge-shaped Hypergranulosis	Characteristic	May be present	May be present
Focal Parakeratosis	Not characteristic	May be present	May be present
Focal Interruption of Granular Layer	Not characteristic	May be present	May be present
Sawtoothing of Rete Ridges	Characteristic	May be present	May be present
Focal Spongiosis	Not characteristic	May be present	May be present
Necrotic Keratinocytes Number At All Levels of Epidermis	Fewer than LDE, V-ILE Not characteristic.	Larger number [93]. May cluster. May be seen at all levels.	May be more than seen in ILP. This feature was seen in 5/15 (33%) cases reported herein.
Location of Cytoid Bodies in Epidermis	Lower spinous layer	Lower spinous layer; may also be seen in upper spinous, granular, cornified layers.	Lower spinous layer; may also be seen in upper spinous, granular, cornified layers.
Lymphocytic Infiltrate Throughout Epidermis	Not characteristic	May be present	Present in 6/15 (40%) cases in series reported herein.
Location of Lymphocytic Infiltrate in Dermis	Superficial (papillary)	Superficial and may extend deeper	Superficial and may extend deeper
Deep Perivascular Infiltrate	Rarely seen	May be present	Often present in lichen striatus Occasionally seen in other lichenoid reactions
Focally lichenoid or patchy infiltrate	Dense band-like infiltrate is characteristic.	May be present	Present in 7/15 (47%) of cases reported herein.
Perieccrine or Periadnexal Infiltrate	Not characteristic. May be seen in adnexotrophic variants.	Uncommon	Often seen in vaccine-induced LS.
Eosinophils in Infiltrate	Not characteristic	May be present. Found in 2/15 cases (13.4%) in study directly comparing LDE to ILP [93].	May be present. Found in 1/15 (6.7%) cases reported herein.
Plasma Cells in Infiltrate	Not characteristic	May be present	May be present
Granuloma Formation	Not characteristic	May rarely be present	Seen in 1/15 cases in series reported herein.

**Table 5 vaccines-11-00438-t005:** Vaccine-Induced Lichenoid Eruptions with a Blaschkoid Distribution.

Author/YEAR Diagnosis	Age M/F Injection Site	Location of Eruption	Type of Vaccine Manufacturer	Interval	Treatment	Clinical Course
Sato [88] 2010 Case 1 Linear Lichen Planus	71 F L Arm	L Buttock + L Thigh/Leg	Influenza Kaketsuken Astelas	7 days after vaccination	Topical Corticosteroids	Resolved with slight pigmentation. Recurrence 1 week after influenza vaccination following year.
Sato 2010 Case 2 Linear Lichen Planus	70F Site NR	L Leg	Influenza NR	2 weeks after vaccination	Topical Corticosteroids	Resolved completely after 6 months.
Hardy [89] 2019 Linear Lichen Planus	43 M L Arm	L Side of Trunk	Influenza NR	8–10 days after vaccination	Topical Corticosteroids	Progression noticed at 1 year follow-up.
Garcia-Martinez [90] 2015 Blaschkoid LP	50 F L Arm	Left Side of Body Leg, Lumbar, Abdomen, Arm	Influenza NR	2 weeks after vaccination	Topical Corticosteroid Oral Corticosteroids	Resolved over 6 months with residual hyperpigmentation.
Bellina [60] 2021 Lichen striatus	42 F R Deltoid	R Arm	COVID-19 Pfizer-BioN Tech	3 days after V2	Tacrolimus Ointment 0.1%	NR Not reported
Kato [61] 2022 Linear Lichen Planus	57 F L Deltoid	L Arm	COVID-19 Pfizer-BioN Tech	2 weeks after third dose	Topical Corticosteroids Short course of Oral Corticosteroids	Improvement with mild residual hyperpigmentation.
Rovira-Lopez [62] 2022 BLAISE	46 F L Deltoid	L-Sided Eruption on Trunk	COVID-19 Pfizer-BioN Tech	“A few days” after V1 No flare after V2	Oral Corticosteroids and potent Topical Corticosteroids	Resolved completely with residual hyperpigmentation.
Case 1 BLAISE	38 F L Deltoid	L Arm	COVID-19 Pfizer-BioN Tech	2 weeks after V2	TAC Ointment 0.1% Oral AH	Resolved with residual mild hyperpigmentation.
Case 2 BLAISE	54 F L Deltoid	Bilateral Legs	COVID-19 Pfizer-BioN Tech	8 days after V1 2 weeks after V2	TAC Ointment 0.1% Prednisone	Resolved
Case 3 BLAISE, LP-like	69 M L Deltoid	L-Sided Trunk	COVID-19 Moderna	1 week after V2	TAC Ointment 0.1%	Resolved with residual hyperpigmentation.
Case 4 BLAISE, LS-like	42 F L Deltoid	L Side of Neck	COVID-19 Pfizer-BioN Tech	10 days after V2	TAC Ointment 0.1%	Resolved with residual hyperpigmentation.

L, left; R, right; NR, not reported; V1, first vaccination; V2, second vaccination; TAC, triamcinolone; AH, antihistamine; LP-lilke, lichen planus-like; LS-like, lichen striatus-like; BLAISE, blaschkolinear acquired inflammatory skin eruption.

## Data Availability

Data supporting reported results have been included in the Tables in this manuscript.

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
