# Peer review of "COVID-19 Vaccine-Induced Lichenoid Eruptions—Clinical and Histopathologic Spectrum in a Case Series of Fifteen Patients with Review of the Literature"

_vaccines, 2023, doi:10.3390/vaccines11020438_

Round 1

Reviewer 1 Report

Sapadin and colleagues provide a comprehensive review on lichenoid skin eruptions that evolve in the context of vaccination (V-ILE) against SARS-CoV-2. Furthermore, they present a case series of 15 patients who experienced V-ILE upon vaccination.

Meritoriously, the authors provide a systematic overview on lichen planus and related lichenoid eruptions including BLAISE, Blaschkitis and lichen striatus. After defining their clinical classification they present 15 own cases in detail and provide clinical images and (immuno)histological micrographs of mainly high quality. Then, they present an overview on 31 cases of SARS-CoV-2 V-ILE cases published in the literature. They conclude that V-ILE may not be fully appreciated and is probably underreported. They state that the immunopathology of V-ILE is widely opaque and that a better understanding would offer new insights into autoimmune phenomena related to SARS-CoV-2 vaccination.

Unfortunately, however, the manuscript suffers from excess length which is partly due to redundancies. I believe that this manuscript would benefit from a more concise presentation of the relevant data. I do not think that it is necessary to include so many clinical images and to present data extensively both in the text and hard-to-read tables.

Comments in detail:

__Figure 2: provide source of this Figure (unless it has been drawn by the authors)

__ Line 57: “or or”: delete one “or”

__Line 201: the context of this statement is not clear

__Figure 5: why is this Figure presented with a black frame?

__Figure 6: please also show an overview that includes the epidermis of this HE staining at a  lower magnification

__Figure 8: the arrow referred to in the legend is missing in the micrograph

__Figure 12, Legend, line 445: “immunostaining using CD123” is not correct; you used an antibody to visualize CD123.

__Table 1: the lay-out of this and some other tables (e.g. Table 5) is horrible – this needs to be improved considerably

__Tables and Figures: explain all abbreviations in footnotes or legends. What does, e. g. “ILTAC” (Figure 20) and “PIH” (Figure 21) mean, what is NR, V-REPP, etc. (Table 4)?

__Figures 18 and 19: omit percentages (yellow numbers)

__lines 602-605: the numbers provided in the text are not compatible with the numbers shown in Figure 20 – please check.

__Table 3: “mean age … 65”: what is the source of this number, add units (i.e. year) 

__CONCLUSIONS section (lines 504ff): The order of thematic paragraphs appears to be somehow confusing. Why, e. g. are the data of the literature search are not presented in the Results Section but in the Conclusions Section? Within the Conclusions Sections, the comparison of own cases with the literature cases is provided earlier than the results of the literature search – why? The discussion on etiology, BLAISE, immunopathogenesis is quite erratically intermingled within sections of descriptive data presentation – I am missing the “golden thread” especially in the Conclusions Section.

Reviewer 2 Report

In the manuscript “COVID-19 Vaccine-Induced Lichenoid Eruption – Clinical and Histopathologic Spectrum in a Case Series of Fifteen Patients with Review of Literature” authors describe a case series and proceed to a literature review. The subject is interesting and relevant but it is necessary some improvement in the manuscript. The description of each case is exhaustive, it may be shortened in my opinion

-          Tables appears as 1,2, 3…, and in the text as I, II, III…, please choose a uniform nomination

-          Fig 4B has not been referred in the text

-          Table 2 is absent, I suppose that it has been mixed with table 1

-          Figure 11 – please nominate the A, B and C

-          page 18, line 431: fig 13 is not the figure, probably the correct is fig 12 B and C

-          conclusions of the case series: I suggest the conclusion at the end of the manuscript,

-          Fig 15 to 22 has not been referred in the text

-          Fig 25 – please correct the subtitles C and D

-          Fig 26 – please correct subtitles, A B C and D

-          Fig 25 and 26 may be put together with the respective case pictures, at the beginning of the manuscript

Round 2

Reviewer 2 Report

The manuscrip "COVID-19 Vaccine-Induced Lichenoid Eruptions - Clinical and Histopathologic Spectrum in a Case Series of Fifteen Patients with Review of the Literature" is well written and provides a comprehensive review of literature.